# Functional Implications of Dynamic Structures of Intrinsically Disordered Proteins Revealed by High-Speed AFM Imaging

**DOI:** 10.3390/biom12121876

**Published:** 2022-12-14

**Authors:** Toshio Ando

**Affiliations:** Nano Life Science Institute, Kanazawa University, Kanazawa 920-1192, Japan; tando@staff.kanazawa-u.ac.jp

**Keywords:** intrinsically disordered proteins, IDPs, intrinsically disordered regions, IDRs, high-speed AFM, dynamic structure, bioimaging

## Abstract

The unique functions of intrinsically disordered proteins (IDPs) depend on their dynamic protean structure that often eludes analysis. High-speed atomic force microscopy (HS-AFM) can conduct this difficult analysis by directly visualizing individual IDP molecules in dynamic motion at sub-molecular resolution. After brief descriptions of the microscopy technique, this review first shows that the intermittent tip–sample contact does not alter the dynamic structure of IDPs and then describes how the number of amino acids contained in a fully disordered region can be estimated from its HS-AFM images. Next, the functional relevance of a dumbbell-like structure that has often been observed on IDPs is discussed. Finally, the dynamic structural information of two measles virus IDPs acquired from their HS-AFM and NMR analyses is described together with its functional implications.

## 1. Introduction

In physiological conditions, intrinsically disordered proteins (IDPs) are devoid of a well-defined three-dimensional structure entirely or partly. And yet, they can carry out biological functions and often work in the regulation of transcription and translation and in cellular signal transduction [1,2]. Therefore, IDPs are more abundant in eukaryotic cells than in prokaryotic cells. Interestingly, IDPs are relatively abundant in viruses [3,4]; Sonia Longhi and colleagues first found a viral IDP in the measles virus [5]. This abundance relative to the total number of viral proteins is possibly because viruses have highjacked the cellular machinery of their hosts. The intrinsically disordered regions (IDRs) contained in IDPs are highly protean. The residues in an IDR responsible for weak intramolecular interactions are widely distributed in space and time. Therefore, IDRs can dynamically sample multiple conformations, including fully disordered tail-like structures, loosely folded conformations, and incompletely or fully folded secondary structures. The residues responsible for target binding are also distributed in space and time, enabling an IDP to bind multiple targets by structural adaptation, resulting in different folded structures by coupled folding, as exemplified in the binding of an IDP (p53) to different structured proteins [6]. Therefore, IDPs can be multifunctional. The coupled folding of an IDR induced by binding to structured proteins [7] may imply that strong binding requires folding of the IDR. However, dimerization or target binding of IDRs without folding has been observed [8,9]. Their dissociation constants (*K*_d_) are in the order of 10 μM. A recent study has suggested that two highly disordered IDPs can bind each other with much higher affinity while retaining their structural disorder and highly dynamic character [10,11]. Moreover, the flexible and adaptable nature of IDRs bestows on them high sensitivity to environmental changes and post-translational modifications. That is, the structure of IDPs and their inter-molecular interactions can be altered by minute changes in the intracellular environment and post-translational modifications. The biological importance of such high sensitivity is typically exemplified by the formation of liquid droplet-like condensates (membrane-less organelles) by liquid-liquid phase separation (LLPS) [12,13,14,15,16]. Thus, the flexible, dynamic structural nature of IDRs is the basis for IDPs’ distinct functions and functional mechanisms that cannot be achieved by structured proteins.

As such, elucidating the conformational ensemble and dynamics of IDPs is essential for understanding their functional mechanisms. However, various techniques developed and used for studying the structure of structured proteins are often not instrumental in the structural analysis of IDPs. The two major structural biological techniques, X-ray crystallography and cryogenic electron microscopy (cryo-EM), do not have practical applications for IDPs because IDRs hamper crystallization and are too thin to be visualized by cryo-EM. Two ensemble-averaging methods, small-angle X-ray scattering (SAXS) [17,18] and solution NMR spectroscopy [19], have mainly been used for structural analysis of IDPs (for their approximate performances, see Table 1). Various SAXS-derived data, such as radius of gyration (*R*g) and Kratky plot, provide information about the overall structural properties of IDPs. However, distinguishing individual structures of an IDP coexisting in solution and quantifying their populations require modeling of structural ensembles. For modeling, several methods have been developed, such as the ensemble optimization method (EOM) [20,21], ILBOMD [22], and MultiFoXS [23]. However, the resolution of SAXS is low (usually ~5 nm) [18,24], and the unique determination of multiple structures and their populations from SAXS data alone is difficult. Therefore, SAXS analysis is usually combined with molecular dynamics simulations or NMR analysis. In contrast to SAXS, NMR cannot determine the overall structure of IDPs. Instead, NMR determines high-resolution local structures of folded regions, transiently appearing structured regions, and residues interacting with targets using two-dimensional ^1^H-^15^N/^1^H-^13^C heteronuclear single quantum correlation spectra. This also means that NMR can specify disordered regions in an IDP that do not interact with other residues. Importantly, NMR allows model-based population analysis for multiple conformational states. A recent study proposed an improved NMR method for ensemble analysis based on solvent paramagnetic relaxation enhancement, enabling accurate, high-sensitive detection of low populations of residual structure [25]. However, NMR is still poor at distinguishing fully disordered and loosely folded structures. Loosely folded structures are often identified wrongly as fully disordered. Besides, there is an upper limit of analyzable molecular weight (~40 kDa). For larger IDPs, their deletion constructs with smaller molecular weights have to be subjected to NMR analysis. However, the analysis in this way does not necessarily guarantee that the revealed structures of segments are identical to those in the full-length protein. This is because intramolecular interactions within an IDP facilitated by a flexible IDR can alter the IDP structure, as has been demonstrated for Atg13 [26,27]. Moreover, it is hard to determine the rates of transitions between different conformational states that occur within >~100 ms, although the fast motion of individual residues can be detected by NMR relaxation analysis without information on structural coordinates [28].

As a complement to the limited capability of these ensemble-averaging methods, we need a single-molecule method that can observe the dynamic protean structure of individual IDP molecules, even at a spatial resolution lower than NMR. Conventional atomic force microscopy (AFM) was previously used to characterize IDPs. However, its target was aggregated assemblies of an IDP or constructs of structured domains within IDPs, such as the nonstructural protein 3 (a protease with folded structure, unlike the name of this protein) from the hepatitis C virus [29] and the DNA binding domain of p53 [30]. Crystal structures of these domains have been obtained. Single molecules of IDPs with disordered regions of >20 residues have never been successfully imaged with conventional AFM. Therefore, we have explored the potential of high-speed AFM (HS-AFM) in the dynamic structural analysis of IDPs. This relatively young microscopy has been successfully used to visualize the dynamic behavior of structured proteins during their functional activity at 2–3 nm lateral and ~0.1 nm vertical spatial resolution and at ~100 ms temporal resolution under near-physiological conditions, without any chemical labeling (Table 1), as demonstrated for a variety of structured proteins [31,32,33,34,35]. IDPs have also been successfully visualized by HS-AFM: e.g., facilitates chromatin transcription (FACT) protein [36,37], the archaeal Hef protein [38], and the bacterial flagellar hook-length control protein FliK [39]. These studies have demonstrated the capability of HS-AFM to visualize the thin structure of IDRs in dynamic motion. Importantly, as demonstrated in a recent study [40], HS-AFM can determine approximately the number of amino acids contained in a fully disordered IDR. Thus, HS-AFM can delineate and quantify the dynamic structure of IDPs, as demonstrated for several IDPs, including viral proteins [40]. Moreover, function-related phenomena of IDPs have also been visualized in recent HS-AFM studies: e.g., ion-dependent structural transitions of a stator ring of the bacterial flagellar motor between unfolded and folded states [41], amyloid fibril formation from *α*-synuclein [42] and Sup35 [43], the assembly process of translation factors around the ribosomal stalk complex [44] and the formation of the droplet-like pre-autophagosomal structure (PAS) via LLPS [14]. In this review, some recent HS-AFM studies on IDPs, including two measles virus IDPs and those with a dumbbell-like shape, are described after a brief introduction of HS-AFM and verification of the absence of AFM-tip disturbance to IDP structure.

## 2. HS-AFM Imaging in Amplitude Modulation Mode

In the amplitude modulation mode, the AFM cantilever oscillates in the Z-direction so that the sharp tip attached to the free end of the cantilever contacts with the sample surface intermittently. This intermittent contact can avoid sample dragging during lateral scanning of the sample stage relative to the tip. Upon tip-sample contact, the cantilever oscillation amplitude is reduced. To maintain the contact (tapping) force constant, feedback control moves the sample stage in the Z-direction so that the oscillation amplitude returns to its set point value. As a result, the motion of the sample stage traces the sample surface (the Z-motion direction is opposite to the sample height). Therefore, we can construct the sample surface topography using a computer from the signal that drives the sample stage in the Z-direction. In our HS-AFM system, all instrumental components, including the cantilever, the scanner, and all electronics, are optimized for highly frequent tip-sample contact, fast conversion of the cantilever deflection signal to the amplitude signal, fast data sampling, fast scanning of the sample stage, or fast feedback control. To minimize the tip-sample contact strength, the free oscillation amplitude *A*_0_ of a short cantilever (7–10 μm) with a spring constant of 0.1–0.2 N/m is set at 1–2 nm, and its set point is set at 0.85–0.9 × *A*_0_, resulting in the cantilever’s oscillation energy loss per tap is only a few *k*_B_*T* on average (*k*_B_, Boltzmann constant; *T*, the temperature in kelvin). The feedback control speed depends on the ratio of *A*_0_ to the sample height *h*_s_; i.e., the control speed increases with increasing *A*_0_/*h*_s_. Since the height of IDPs is generally small (the mean height of a fully disordered IDR is about 0.5 nm), the imaging speed can be increased up to 50 fps for many IDPs. Very recent improvements to our HS-AFM system have enabled ~70 fps even for structured proteins with *h*_s_ < 25 nm, although this improved system has not yet been used for IDPs. For HS-AFM imaging of biological molecules, they have to be placed on a substrate surface (Figure 1a). HS-AFM imaging of IDPs requires the surface to be hydrophilic and flat at the atomic level. Practically, bare mica that meets these requirements is probably the only choice. In general, IDRs only weakly attach to mica via electrostatic and van der Waals interactions and, therefore, exhibit fast Brownian motion on mica. When an IDP is disordered over the entire length, it diffuses too fast to be imaged even at 50 fps. In such a case, we need to fuse a folded protein (e.g., GFP) to the N- or C-terminus of the IDP as an anchor onto mica. More details of practical issues in HS-AFM imaging of IDPs are described in [45]. For more comprehensive descriptions of the fundamentals, techniques, and biological applications of HS-AFM, a book is available [46]. HS-AFM systems have already been commercialized by three manufacturers (RIBM, Bruker-JPK, and Oxford Instruments-Asylum). Short cantilevers with a spring constant of 0.1–0.2 N/m and a resonant frequency higher than 400 kHz in water are also commercially available from NanoWorld and RIBM.

## 3. No Effect of Tip-Sample Contact on the Structure of IDPs

The tip-sample interaction is much weaker in liquids than in air because of the lack of capillarity forces in liquids [47]. Nevertheless, there still may be a concern that the physical contact between a protein molecule and an oscillating tip in an aqueous solution would have a strong impact on the structure and function of the molecule. The absence of such an impact has been verified for various structured proteins. For example, the HS-AFM images that captured the processive movement of myosin V on actin filaments at 7 fps showed that its movement velocity is identical to that measured by fluorescence microscopy in the same buffer solution condition [31]. The conformational changes of F_1_-ATPase filmed with HS-AFM at 12.5 fps for over 40 s were observed to continue without the changes being debilitated [32]. During this imaging, the molecules were tapped with the AFM tip tens of millions of times. However, the absence of a tip disturbance to the dynamic structure of IDPs cannot be verified straightforwardly because the dynamic conformational changes of IDPs occur thermally and stochastically.

Although the feedback control error is generally small for low height samples with *h*_s_ ≤ *A*_0_, the error becomes larger (and hence the tip-sample interaction force becomes stronger in both X- and Z-directions) with increasing X-scanning velocity. Therefore, the best way to examine the possible tip disturbance to IDPs is to compare the characteristic structural parameters of an IDP quantified from its images filmed at various scanning velocities. Polyglutamine tract-binding protein 1 (PQBP-1) and its deletion construct PQBP-1 (1–214) were subjected to this examination [40]. PQBP-1 has an N-terminal globular domain (1–81) containing a WW domain (48–81) followed by a constantly disordered region (82–265). The three structural parameters of PQBP-1 (1–214) were measured from many images captured at six different imaging rates (6.7–50 fps): the height of the globule (*H*_1_), the height of the disordered region (*H*_2_), and the two-dimensional (2D) end-to-end distance between the N-terminal globule and the C-terminal end (*R*_2D_). The *R*_2D_ was calculated as *R*_2D_ = *D* − *H*_1_/2 − *H*_2_/2, where *D* is the direct distance between the N-terminal globule and the C-terminal end. Note that *R*_2D_ can be measured accurately despite the fact that all pixel data contained in an image are not acquired simultaneously but at different times due to raster scanning of the sample stage during one-frame acquisition. This is because the N-terminal globule is nearly stationary, although the C-terminal IDR moves fast on mica; the positions of the N-terminal globule and the C-terminal end can be considered as detected at the same time. As shown in Figure 1b–d, the mean values of these parameters, (*H*_1_), (*H*_2_), and (*R*_2D_), were independent of the imaging rate. Moreover, the displacement of the C-terminal end of WT PQBP-1 in the fast scan X-direction during a given time interval was measured relative to the N-terminal globule. The displacement was distributed symmetrically around the zero-mean displacement, irrespective of the time interval (Figure 1e). Note that in raster scanning, the sample surface is traced (from –*X* to +*X*) and retraced (from +*X* to –*X*) in the fast scan direction (Figure 1a), but only images captured in trace imaging are used. Therefore, the symmetric distribution of the relative displacement indicates no effect of the tip on the fast motion of the flexible IDR on mica. The time interval independency of the displacement distribution suggests that the shape of the IDR changes much faster than the image acquisition rate. The autocorrelation function of *R*_2D_(*t*) was zero except for *Δt* = 0, even when the HS-AFM images were captured at 50 fps [40]. While the tip intermittently contacts the sample, the mica surface is always in contact with the sample. Therefore, the mica surface may have a non-negligible effect, especially on the highly flexible structure of IDRs. An IDR is confined to a 2D solution space on the mica surface, entailing (*R*_2D_) longer than the mean three-dimensional (3D) end-to-end distance in solution, (*R*_3D_). However, similar confinement can also occur in the crowded cytoplasm, nucleus, and viral capsid.

## 4. Constancy of Flexibility/Rigidity of Fully Disordered Regions

Except for rare cases, chemically denatured polypeptide chains are known to have unique flexibility or rigidity, irrespective of the amino acid sequence and composition [48]. This constancy can be expressed by a power law for the relationship between the number of amino acids *N*_aa_ and the mean radius of gyration, 〈*R*_g_〉: i.e., 〈*R*_g_〉 = 0.1927 nm × *N*_aa_^0.598^ (Figure 2a) [48]. In the 3D worm-like chain model for semi-flexible polymer chains, the mean square of *R*_g_, i.e., 〈*R*_g_^2^〉, can be expressed as a function of the 3D persistence length *P*_3D_ as
(1)$$\langle R_{g}^{2}\rangle =\frac{P_{3D}L}{3}-P_{3D}^{2}+\frac{2P_{3D}^{3}}{L}-\frac{2P_{3D}^{4}}{L^{2}}\times \left[1-\mathrm{Exp}\left(-\frac{L}{P_{3D}}\right)\right]$$
where *L* is the contour length of a polymer chain. In the case of polypeptide chains, *L* can be calculated approximately as *L* = 0.360 nm × (*N*_aa_ − 1); 0.360 nm is the average distance between two adjacent amino acids [49,50]. The 〈*R*_g_〉 data of chemically denatured polypeptide chains approximately followed Equation (1), suggesting that chemically denatured proteins behave like worm-like chains. From this fitting, the value of *P*_3D_ is determined to be 0.976 nm.

In contrast to chemically denatured polypeptides, the SAXS-measured 〈*R*_g_〉 values of 72 IDRs without permanent secondary structures in the native state have been reported not to follow a unique power law [51]. However, detailed structural analyses on some of these IDRs have identified regions that tend to form residual structures [52,53,54,55]. In contrast to SAXS, HS-AFM imaging can detect transiently appearing residual structures, and therefore we can select images of IDRs in the fully disordered state. All the log-log plots of mean 2D end-to-end distance 〈*R*_2D_〉 vs. *N*_aa_ (contained in a fully disordered IDR) measured by the HS-AFM imaging of nine different constructs of IDPs were on a single line representing a power law of 〈*R*_2D_〉 = 1.16 nm × *N*_aa_^0.517^ (Figure 2b) [40]. When worm-like chains freely equilibrate on the surface as in a 2D solution, the relationship between 〈*R*_2D_^2^〉 and the 2D persistence length (*P*_2D_) is expressed as [56,57]
(2)$$\langle R_{2D}^{2}\rangle =4LP_{2D}\left[1-\frac{2P_{2D}}{L}\left\{1-\mathrm{Exp}\left(-\frac{L}{2P_{2D}}\right)\right\}\right].\;$$

Because the nine 〈*R*_2D_〉 data were well fitted to Equation (2), as shown in Figure 2c, these IDRs in the fully disordered state are uniquely characterized by the measured *P*_2D_ value of 1.16 nm. In addition, the fully disordered IDRs are also uniquely characterized by their mean diameter of approximately 0.5 nm. The unique power law enables the determination of *N*_aa_ contained in a fully disordered IDR from its HS-AFM images, as described later.

However, the substrate surface can have an additional effect arising from 2D frictional forces locally exerting from the surface against the fast 2D Brownian motion of a highly flexible IDR chain [58]. In the absence of this effect, 〈*R*_2D_〉 is identical to $2\sqrt{3}$ × 〈*R*_g_〉, where 〈*R*_g_〉 is the mean radius of gyration in 3D solution (not on the substrate surface). The comparison of our 〈*R*_2D_〉 values for the nine different IDP constructs with SAXS-measured 〈*R*_g_〉 values of four PQBP-1 IDR constructs and 10 tau protein constructs without large extended domains [55] quantitatively revealed that IDRs are swelled by the frictional force from mica by a factor of 1.24, i.e., 〈*R*_2D_〉 = 1.24 × $2\sqrt{3}$ × 〈*R*_g_〉. Thus, we could derive a power law for 〈*R*_g_〉 as 〈*R*_g_〉 = 0.248 nm × *N*_aa_^0.532^ and *P*_3D_ = 0.765 nm. In literature, we can find many SAXS-measured 〈*R*_g_〉 data of IDPs that follow this power law [59,60,61,62,63,64,65,66,67,68,69], although the absence of residual structures in these IDPs is unknown. Note that the value of *P*_3D_ (0.765 nm) is smaller than that of chemically denatured polypeptides (0.976 nm), indicating that chemically denatured polypeptides are more extended than fully disordered IDRs. This distinct difference can be arguably explained simply by the fact that denaturants such as urea and guanidinium chloride are good solvents. Therefore, the polypeptide chain is extended because the interaction between the chain backbone and a good solvent is favorable. In contrast, chain-chain interactions are repulsive, although different interpretations may be possible [70].

## 5. IDPs with Dumbbell Shape

IDPs randomly selected for HS-AFM imaging often showed a dumbbell-like shape (i.e., two structured domains are linked via an IDR): e.g., the archaeal Hef (helicase-associated endonuclease for fork-structured DNA) protein [38], bacterial FliK protein [39] and yeast Atg1 [40] (Figure 3a–c). In addition, other methods found various dumbbell-shaped IDPs: e.g., the regulatory subunits of the cAMP-dependent protein kinase [71], cardiac troponin C [72], the DNA repair factor CtIP [73], CAHS (cytoplasmic abundant heat soluble) protein [74], and Ca^2+^-unbound calmodulin [75]. In these IDPs, the N-terminal domain (NTD) and the C-terminal domain (CTD) often possess distinct functions. For example, Hef has a helicase domain at the N-terminus and a nuclease domain at the C-terminus and dimerizes via the nuclease domain. The helicase activity is dramatically stimulated by fork-structured DNAs, while the nuclease specifically cleaves nicked, flapped, and fork-structured DNAs [76,77]. From these activities and phenotype analyses of mutant strains, Hef functions mainly in stalled replication fork repair [78]. In the case of Atg1, the sole kinase in all Atg proteins, the NTD is a kinase domain, while the CTD is the tandem MIT (microtubule interacting and transport) domain that binds the MIT-interacting motif (MIM) in Atg13. The NTD in FliK works as a length ruler of the bacterial flagella hook, while the CTD functions to switch secretion from the hook protein to the filament protein in the type-III flagellar export apparatus [79]. The domain diagrams of these dumbbell-shaped IDPs are shown in Figure 3d. A previous NMR analysis of the FliK^N^ fragment (1–147) indicated a largely unfolded structure, whereas the FliK^C^ fragment (204–370) is structured, except for its N-terminal region 204–252 [80]. The HS-AFM images of dumbbell-shaped FliK showed larger (mean height, 2.7 nm) and smaller (mean height, 2.0 nm) globules connected by a flexible linker (Figure 3c). From HS-AFM images of deletion constructs of FliK, the larger and smaller globules were identified as the NTD and CTD, respectively [39]. However, their size difference is inconsistent with the domain arrangement in FliK (Figure 3d), suggesting that the NTD is loosely folded. There may be two possibilities for the disagreement between HS-AFM and NMR results of the structure of Flik^N^; (i) the loosely folded structure of the FliK^N^ fragment could not be detected but was detected by NMR as unstructured, or (ii) the Flik^N^ fragment is unfolded.

The common structural feature in dumbbell-shaped IDPs must have multiple functional benefits. In the repair of stalled replication fork by Hef, the fork structure has to be altered by the helicase, and DNA segments have to be removed by the nuclease. Although other proteins are also involved in the repair process, it is evident that the helicase and nuclease activities must proceed in harmony. The flexible IDR linker connecting the helicase and nuclease domains can localize these functional sites more efficiently and facilitates their harmonized actions compared to the case where separated helicase and nuclease proteins work independently. In the case of Atg1, its C-terminal MIT1/MIT2 domain can bind the dephosphorylated MIM domain located in the C-terminal disordered region of Atg13. Therefore, starvation-induced dephosphorylation of Atg13 triggers its binding to Atg1 and then recruits Atg17-29-31 and Atg11 to form the Atg1 complex (Atg1-Atg13-Atg17-Atg29-Atg31) that works as a scaffold for PAS formation [14,26]. The Atg13-Atg17 interaction can occur when regions close to the MIM domain are dephosphorylated. Although the reaction timing after PAS or autophagosome formation is not well known, Atg1 kinase phosphorylates Atg13 and Atg11 [27,81], which is considered to dissolve a large complex formed by the PAS complex and other Atg proteins, and regulates selective autophagy [81,82]. When Atg1 phosphorylates the Atg13 IDR by the N-terminal kinase, Atg1 also grasps this IDR using the C-terminal MIT1/MIT2 domain. Thus, the N- and C-terminal domains of Atg1 need to get in close proximity to each other, which can be facilitated by the long flexible linker connecting the two interaction sites.

Another functional benefit of the dumbbell-like structure is that this structure allows and facilitates dynamic interactions between the two distantly separated functional domains. Even when the *K*_d_ of intramolecular interactions between distantly separated domains is large, the effective concentration (*C*_eff_) of one domain around the other can be significantly high, enabling their frequent interactions. The *C*_eff_ can be estimated as a function of $\langle R_{3D}\rangle$, as follows. A fully disordered IDR is considered to behave as a worm-like chain. In this case, the distribution function of *R*_3D_ is given by
(3)$$P\left(R_{3D}\right)dR_{3D}=4\pi R_{3D}^{2}{\left[\frac{3}{2\pi \langle {R_{3D}}^{2}\rangle }\right]}^{3/2}\exp\left[-\frac{3}{2}\frac{R_{3D}{}^{2}}{\langle {R_{3D}}^{2}\rangle }\right]dR_{3D}.$$

The two globular domains linked via such an IDR associate with each other when they approach within a certain range of distance. However, when we consider the actual dimensions of the two globules, *R*_3D_ becomes a complex function of the distance between the two globules. To avoid this complexity, we assume that NTD-CTD association occurs when *R*_3D_ becomes *R*_0_ or shorter (i.e., *R*_3D_ ≤ *R*_0_). In this approximation, the probability of finding the CTD in the volume (*R*_3D_ ≤ *R*_0_) around the NTD is given by
(4)$$p=\int_{0}^{R_{0}}P\left(R_{3D}\right)dR_{3D}$$

Therefore, the *C*_eff_ of the CTD in the volume in close proximity to the NTD is given as
(5)$$C_{\mathrm{eff}}=\alpha \times p\times 10^{-3}/\left(\frac{4\pi }{3}R_{0}^{3}\cdot N_{A}\right)$$
where *N*_A_ is the Avogadro constant and *α* is a steric hindrance effect factor by which the NTD–CTD interaction frequency is reduced. Considering the NTD surface available for CTD binding, the steric hindrance factor is approximately 0.5. Supposing that the IDR contains 100 amino acids, its $\langle R_{3D}\rangle$ value is 7.25 nm, under which Equation (4) provides *p* = 3.56 × 10^−3^. Therefore, the effective concentration of the CTD around the NTD is approximately estimated to be *C*_eff_ = 0.71 mM (the *N*_aa_ dependence of *C*_eff_ is shown in Figure 3e). Thus, the two terminal domains in a dumbbell-shaped IDP can have a large chance to interact with each other even when their *K*_d_ value is in the order of mM.

So far, only 2–3 IDPs with a property of “conformational malleability” have been found [26]. Atg13 is such a rarely found IDP. Although it is not dumbbell-shaped, its N-terminus folded HORMA domain (1–267) interacts with the C-terminus end region of the long IDR (268–740). The HORMA domain binds Atg9, a membrane protein existing on specific cytoplasmic vesicles (Atg9 vesicles) [83]. The C-terminal IDR contains the Atg1 binding site (460–521) and Atg17 binding sites (359–389 and 424–436) [27]. The folded HORMA domain becomes largely disordered when the C-terminus is fused to maltose-binding protein (MBP) [26,27]. The MBP-fusion impairs the PAS localization of Atg9 in the cell. These observations indicate that transient NTD-CTD interactions are required for the formation of the stable and functional HORMA structure because the MBP moiety can interfere with NTD-CTD interactions. The stabilized structure of HORMA can be retained for a while even after the transient interaction is completely removed (i.e., malleability). Therefore, only occasional interactions are enough to keep the stable and functional HORMA structure. And yet, the functional role of this conformational malleability is still unknown. Although not yet clearly demonstrated, Flik possibly exhibits conformational malleability. That is, when the Flik^N^-Flik^C^ interaction in *cis* is lost for a certain time duration, the Flik^N^ domain (1–147) becomes fully disordered. When this domain is fully disordered, its contour length becomes ~53 nm, nearly identical to the maintained hook length of the bacterial flagellum (~55 nm), and thus, the Flik^N^ domain can work as a ruler of hook length. Namely, the N-terminal end of FliK binds to the tip of the growing hook, while the disordered FliK^N^ chain is moderately stretched in the narrow channel of the filament. When the hook grows up to ~55 nm, the Flik^N^ chain is fully stretched and, thus, pulls the Flik^C^ domain remaining in the cytoplasm. This pulling force changes the Flik^C^-secretion gate interaction to switch the secretion substrates from early proteins, including the hook protein, to late proteins, including the flagellin, as advocated previously [79,84].

## 6. Measles Virus IDPs

Before describing the HS-AFM study on two IDPs of measles virus, here I summarize their background. Comprehensive reviews on these IDPs and their functional roles are available elsewhere [85,86,87,88]. The measles virus is a nonsegmented, single-stranded, negative-sense RNA virus. Therefore, after virus entry into host cells, mRNAs coding for eight different virus proteins with six genes need to be synthesized from the genome RNA by using its own RNA-dependent RNA polymerase complex comprising the Large (L) protein and its cofactor, the phosphoprotein (P). This synthesis mode is later switched to the processive mode to produce cRNA for genome replication. The genome RNA is encapsidated by the nucleoprotein (N) to form a helical nucleocapsid with a diameter of ~18 nm and a helical pitch of ~6 nm, comprising 12.3 protomers per turn of the helix (Figure 4a), as suggested from the cryo-EM analysis of nucleocapsid-like particles [89]. This ribonucleoprotein complex is the substrate for both transcription and replication; the polymerase complex (L-P) cartwheels on the nucleocapsid template in these processes (Figure 4a). Each N binds six nucleotides, and therefore, the genome length (15,894 nucleotides) is exactly 6 × integer (2649). Monomeric N also exists in the cell as a soluble monomer which is referred to as N^0^, while the assembled form is referred to as N^NUC^. N^0^ requires its chaperone, whose role is played by P and the V protein, which shares the first 231 amino acids with P [88]. The binding of P (and also V) blocks the polymerization of N. The tetrameric P tethers L onto the nucleocapsid template (Figure 4a). Both the N and P proteins contain large, disordered regions (Figure 4b). The N protein consists of the N-terminal structured domain (N_core_: 1–400) and the C-terminal disordered domain (N_TAIL_: 401–525). Note that the aforementioned nucleocapsid-like particles are formed without N_TAIL_; in the presence of N_TAIL_, the nucleocapsid is very flexible. The P protein consists of the N-terminal domain containing a large, disordered region (PNT: 1–230) and the C-terminal domain (PCT: 231–507) (Figure 4b). Transcription requires only PCT, whereas genome replication also requires PNT. The N-terminal PNT moiety (P_N37_: 1–37) binds N^0^, while the PCT region (304–376), referred to as PMD, is responsible for coiled-coil oligomerization of P and binding to L, and the C-terminal end region (459–507) referred to as the X domain (XD) is responsible for binding to N_TAIL_ (the segment P_Loop_ connecting PMD and XD is disordered). Thus, the formation of the N^0^-P complex is mediated by the dual PNT-N_CORE_ and PCT-N_TAIL_ interaction. In a different description of the domain organization of P, the region 1–304, larger than PNT, is referred to as P_TAIL_, which is mostly disordered except for P_N37_ [88].

There are fundamental questions to be solved about the measles viral transcription and replication processes. For example, (i) how does the polymerase switch between transcriptive and replicative modes? (ii) how are viral and cellular RNAs discriminated by the N protein? (iii) how can the L-P complex access the RNA which winds around the groove of the helical polymer of N protein? and (iv) why does the N^0^-P complex need to be formed before the N protein polymerizes? Various attempts to answer these questions and others have been carried out based on the structural features of N and P and their interactions. Some of the mechanistic proposals that answer these questions are summarized here. For more details, see reviews [85,86,87,88]. For instance, it has been proposed that the mode switch between transcription and replication is due to a conformational change of the nucleocapsid template caused by the folding of N_TAIL_ induced by the interaction with XD [85]. However, a completely different mechanism has also been proposed for a negative-sense RNA virus (influenza virus). That is, there may be no mode switch, but nascent cRNA is degraded by host cell nucleases unless it is stabilized by newly synthesized N and P [90]. N_TAIL_-XD binding is also suggested to assist L in remaining in contact with the nucleocapsid template during successive synthesis of mRNAs [91]. The binding of P_N37_ (and also V_N37_) to N^0^ keeps N^0^ in an open conformation ready to grasp viral RNA and is suggested to avoid non-specific binding of N to cellular RNA [92]. N_TAIL_ is suggested to loosen the helically assembled ribonucleoprotein complex, facilitating the transcription and replication by L-P [93].

## 7. HS-AFM Study on N_TAIL_

Prior to the HS-AFM study on N_TAIL_ (401–525) [40], its structural features have already been well characterized quantitatively by various methods. The two regions, called Box1 (401–420) and Box2 (486–502), were previously shown to contain a linear motif with *α*-helical propensity (*α*-MoRE) (Figure 4c) [93,94] undergoing transitions among completely unfolded and folded helical conformations [95,96,97,98,99]. The NMR-detected unfolded state of Box2 is short-lived (μs order) [86]. The binding of Box2 to XD in a triple *α*-helical bundle is suggested to form a quadruple *α*-helical bundle [100]. Box1 is located in the groove of N^NUC^ (Figure 4c) [89,93]. In addition to Box1 and Box2, the region (517–525) is called Box3 because it is suggested to bind XD [101]. However, Box3 is not considered *α*-MoRE because the secondary structure stabilizer, 2,2,2-trifluoroethanol (TFE), does not promote *α*-helical folding within Box3 [101], unlike Box1 and Box2. Box3 possibly remains disordered even after binding to XD. Thus, the regions containing 88 amino acids in the two IDRs (421–485 and 503–525) are considered to be fully disordered. Since most of these structural features of N_TAIL_ are already established, the HS-AFM imaging of N_TAIL_ constructs aimed to examine whether HS-AFM could faithfully detect these known structural features of N_TAIL_ and to determine dynamic structural transition rates that could not have been analyzed with other methods. As already included in Figure 2b, 〈*R*_2D_〉 = 11.6 ± 4.4 nm measured with the HS-AFM imaging of N_TAIL_-GFP fusion (Figure 5a) approximately followed the power law, 〈*R*_2D_〉 = 1.16 nm × *N*_aa_^0.517^ (GFP was used to impede the otherwise very fast diffusion of N_TAIL_ on mica). In this fusion construct, Box1 was always visible, whereas Box2 only occasionally appeared because it is positioned close to the large GFP moiety. The height distribution of Box1 was well fitted to a double Gaussian function with peaks at 0.8 nm and 1.1 nm (Figure 5b). The former is larger than the height of fully disordered regions (~0.5 nm), while the latter is identical to the diameter of a single *α*-helix. The area ratio of the two Gaussian components (*K*_e_ = higher state/lower state) was 0.28, indicating a propensity of Box1 to take partially disordered conformations. To observe dynamic structural transitions of Box2, N_TAIL_ was fused to thioredoxin (Trx) at the N-terminus of N_TAIL_ (Figure 5c). The height distribution of Box2 was also well fitted to a double Gaussian function with peaks at 0.8 nm and 1.1 nm (Figure 5d). However, Box2 tends to form an *α*-helix compared to Box1, judging from its larger *K*_e_ value, 1.91. From these observations, we conclude N_TAIL_ is fully disordered except for Box1 and Box2. Since the measured *R*_2D_ values are widely distributed due to the highly flexible nature of the disordered regions, changes of *R*_2D_ resulting from the structural transitions of small Box1 and Box2 are masked by this wide distribution of *R*_2D_ and their preferred tendencies to take partially disordered and fully folded states, respectively. Nevertheless, the time-varying *R*_2D_ values are well correlated with the time-varying height values of Box1 and Box2, suggesting that the IDR length partially changes due to the transitions in these globules between the partially and fully folded states. One of the advantages of HS-AFM imaging of IDPs over other methods is that we can directly observe structural transitions and, therefore, determine the transition rates unless they are faster than ~50 s^−1^. To determine the transition rates of Box1 and Box2, the autocorrelation functions *G*(*τ*) of their time-series height data *H*(*t*) were first calculated (Figure 5e,f) as
(6)$$G\left(\tau \right)\equiv \sum_{n=1}^{N-h}\left[H\left(n\Delta t\right)-\hat{\mu }\right]\left[H\left(\left(n+h\right)\Delta t\right)-\hat{\mu }\right]/\sum_{n=1}^{N}{\left[H\left(n\Delta t\right)-\hat{\mu }\right]}^{2}$$
where *τ* ≡ *h*Δ*t* (*h* = 0, 1, 2, 3, …, *N*–1) and $\hat{\mu }$ is the mean value of *H*(*n*Δ*t*) (*n* = 1, 2, 3, …, *N*). When the order-to-disorder and disorder-to-order transition rates are expressed by *k*_OD_ and *k*_DO_, respectively, the *G*(*τ*) relaxation rate denoted by *λ* is expressed as *λ* = *k*_OD_ + *k*_DO_. From the relationship, *K*_e_ = *k*_DO_/*k*_OD_, *k*_OD_ and *k*_DO_ can be expressed as functions of *λ* and *K*_e_ as *k*_OD_ = *λ*/(1 + *K*_e_) and *k*_DO_ = *λK*_e_/(1 + *K*_e_). In this way, the rate constants could be determined as *k*_OD_ = 5.7 s^−1^ and *k*_DO_ = 1.6 s^−1^ for Box1 and *k*_OD_ = 1.1 s^−1^ and *k*_DO_ = 2.0 s^−1^ for Box2. Thus, the dynamic structural features of N_TAIL_ are quantitatively delineated, as shown in Figure 5g. In both Box1 and Box2, the structural transitions are relatively slow, comparable to the elongation rate of the L-P polymerase (three nucleotides/s) [102]. Although this rate comparability may be just a coincidence, N_TAIL_ has to change its structure in concert with the motion of the L-P polymerase cartwheeling on the nucleocapsid template. For example, Box2 has to change its affinity for P (at XD) during this L-P polymerase motion. In contrast, Box1 has to be either disordered or folded (but likely the former) to loosen the helical nucleocapsid structure by interfering with the stacking of turns (see Figure 4a,c) [93,103] so that the polymerase reaction is facilitated.

## 8. HS-AFM Study on PNT

When we started the HS-AFM imaging of PNT in 2012, nothing was known about its structure except for the resistance to proteolysis of the N-terminal 27–99 region of PNT in the presence of TFE [104]. In 2016, a crystal structure was revealed for a fused protein complex, P_1–48_-N^0^_21–408_, that is devoid of the disordered regions of P and N [105]. This chimera protein is dimerized in the crystal, and P_1–48_ forms two *α*-helices (longer *α*1 and shorter *α*2) connected with a short loop. The *α*1 and *α*2 bridge two adjacent N^0^_21–408_ protomers, although this arrangement in the chimera proteins may not occur in the physiological case. It is also uncertain whether the N-terminal region, P (1–48), forms *α*-helices without N^0^. In 2018, the Blackledge group reported the NMR structures of P_TAIL_ (1–304) alone and the complex between the full length of N and P_TAIL_ [106]. In the NMR structure of P_TAIL_ alone, four regions were identified to have propensities to form *α*-helices and thus named *α*1/2 (1–37), *α*3 (87–93), and *α*4 (189–198). These three regions are linked by highly flexible IDRs. The IDR segment (125–170) that links *α*3 and *α*4 is rich in acidic residues and devoid of basic residues and, therefore, referred to as the acidic loop. In the N^0^-P_TAIL_ complex structure, a region containing *α*4 (unstructured ^181^DVETA^185^ termed δ + *α*4) was identified as an additional N interaction site. This interaction occurs independently of the presence of *α*1/2. Its counterpart in N responsible for this additional interaction was identified to be a contiguous region 96–127 in N_CORE_. This interaction occurs transiently with a time constant in the order of ms and *K*_d_ = 0.6 mM. Although this *K*_d_ is large, the effective concentration *C*_eff_ of δ*α*4 around the N_CORE_ can be significantly enhanced by the *α*1/2-N_CORE_ and N_TAIL_-XD associations and the high flexibility of the linker segments, as mentioned in Section 5. An essential role of this δ*α*4–N_CORE_ interaction in the polymerase reaction was shown by a transcription assay. Since *α*3 and the acidic loop are free from the other part of P_TAIL_ and not bound to N_CORE_, the highly flexible motion of the long flexible segment between *α*1/2 and δ*α*4 dynamically occupies a large space around N_CORE_. This large space occupation potentially inhibits the interaction of RNA, N monomers, and other proteins with N_CORE_, as discussed in this study [106]. As demonstrated in the HS-AFM study on a yeast prion protein Sup35, the highly flexible motion of long IDRs can have such a repelling effect [43]. However, despite these structural details revealed by the NMR analysis, the resistance of the N-terminal 27–99 region of PNT to proteolysis cannot be rationalized from the NMR data, suggesting the existence of a large-scale structure undetectable by NMR.

As described below, the combination of the NMR structure of P_TAIL_ and results from the HS-AFM imaging of PNT (1–229)-GFP fusion enriched our understanding of the dynamic structure of PNT and its functional relevance. The HS-AFM images of PNT-GFP (Figure 6a) showed an N-terminal globule connected to the GFP moiety via a flexible chain undergoing length changes. In addition, a small globule close to GFP could be occasionally detected, probably corresponding to *α*4. The mean height of the N-terminal globule remained constant at 1.1 nm (Figure 6b) while its lateral dimensions varied. The height of 1.1 nm corresponds to the diameter of a single *α*-helix. Therefore, the N-terminal globule contains *α*1/2 (and possibly also *α*3) as the highest particle and other regions lower than *α*1/2. The *R*_2D_ value of the flexible chain connecting the GFP and the N-terminal small globule is distributed widely. Its distribution was well fitted to a double Gaussian function with peaks at 8.9 and 14.3 nm (Figure 6c). The mean height of the flexible chain was 0.4–0.5 nm in both metastable shorter and longer states. This suggests that the flexible chain is always fully disordered in both metastable states, except for the occasionally appearing *α*4 (*α*3 did not appear in the HS-AFM images as a distinct region). From the power law, 〈*R*_2D_〉 = 1.16 nm × *N*_aa_^0.517^, the *N*_aa_ contained in the shorter and longer chains were estimated to be 51 and 128, respectively. The presence of small *α*3 and *α*4 regions that may be undergoing structural transitions does not much affect this estimation. From these *N*_aa_ values, the number of amino acids contained in the N-terminal small globule was approximately estimated to be 178 and 101 for the shorter and longer chain states, respectively. From the number 101, the N-terminal small globule always contains *α*1/2 (1–37), *α*3 (87–93), and an IDR segment (38–86) connecting *α*1/2 and *α*3. Therefore, these regions always form a compact structure, explaining no appearance of *α*3 as a distinct region in the HS-AFM images. Importantly, this compact structure also explains the resistance to proteolysis of the N-terminal segment (27–99) containing the fully disordered IDR segment (38–86). The number 178 indicates that the structural transitions of PNT occur on a large scale, mainly by association and dissociation between the N-terminal compact structure and the acidic loop (125–170). From the degree of order propensity, *K*_e_ = 3.73, the larger compact structure, including the acidic loop, is formed more frequently than the smaller compact structure without the acidic loop. The *k*_OD_ and *k*_DO_ values of the observed global structural transitions were estimated as *k*_OD_ = 0.5 s^−1^ and *k*_DO_ = 2.0 s^−1^ from the correlation function of time-series *R*_2D_ data (Figure 6d) and the *K*_e_ value. Thus, the dynamic structural features of PNT are quantitatively delineated, as shown in Figure 6e. These rates are again comparable to the elongation rate of the L-P polymerase. In the larger compact form, δ*α*4 cannot reach its interaction site on N_CORE_. *α*1/2 also may not be available for N_CORE_ binding. Since the δ*α*4-N_CORE_ interaction is essential for RNA transcription, the docking and undocking of the acidic loop and the N-terminal structural unit (i.e., the smaller compact structure) must be regulated at certain steps in the transcription process. The cartwheeling L-P around the nucleocapsid may pull the acidic loop for its undocking, enabling smooth transcription. Since the N-terminal smaller and larger compact structures are not detected in the NMR analysis, some of the structural and dynamic features in the N^0^-P complex model constructed from NMR data have to be amended. However, this complex also needs to be investigated by using HS-AFM.

## 9. Outlook

HS-AFM can visualize IDPs at the single molecule level and identify the structural states of their IDRs (e.g., fully disordered, partially folded, and fully folded states) and their dynamic transitions quantitatively. Importantly, the number of amino acids contained in a fully disordered IDR can be approximately estimated. In contrast, the ensemble NMR spectroscopy methods are good at the atomistic determination of local structures and interaction sites in IDPs but poor at detecting large-scale structures and identifying loosely folded regions. Thus, HS-AFM and NMR mutually complement each other. As such, the combination of these techniques can enrich our understanding of the dynamic structure of IDPs, as demonstrated in the studies of measles viral P protein. This review mainly highlighted the structure and dynamics of IDPs revealed by HS-AFM and their functional implications. However, a few HS-AFM studies have recently been carried out to observe the functional phenomena of IDPs, such as amyloid fibril formation and dissolution and the formation of liquid droplet-like condensates via LLPS. This new trend in HS-AFM applications will be expanded in the near future because tools to analyze the functional phenomena of IDPs are limited, while IDP-involved biological phenomena and diseases are emerging in succession. In this regard, the HS-AFM studies on measles virus IDPs also should be expanded to visualize dynamic processes, including N-P interactions, polymerization of N into N^NUC^, and even RNA synthesis. This line of studies may contribute to the development of new drugs against measles; although the measles vaccine is already available, many people still suffer and die from measles.

## Figures and Tables

**Figure 1 biomolecules-12-01876-f001:**
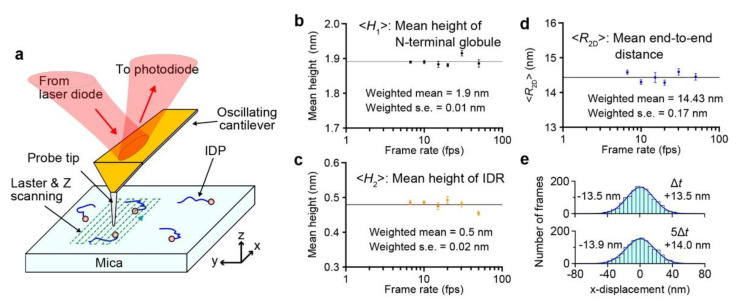
Absence of tip disturbance to the structure of IDPs. (**a**) Schematic showing HS-AFM imaging of IDP molecules placed on mica surface. (**b**–**d**) Molecular features of PQBP-1 (1–214) measured by HS-AFM at various imaging rates (6.7–50 fps): (**b**) Mean height of N-terminal globule, (**c**) mean height of IDR, and (**d**) mean 2D end-to-end distance of IDR. (**e**) Distributions of C-terminal end displacements in the X-direction were observed for WT PQBP-1 during time periods of Δ*t* and 5Δ*t*. Δ*t* is one frame acquisition time (42 ms). The numbers shown on each panel are the mean displacements in the +*X* and −*X* directions. Note that the IDR within PQBP-1 is constantly disordered.

**Figure 2 biomolecules-12-01876-f002:**
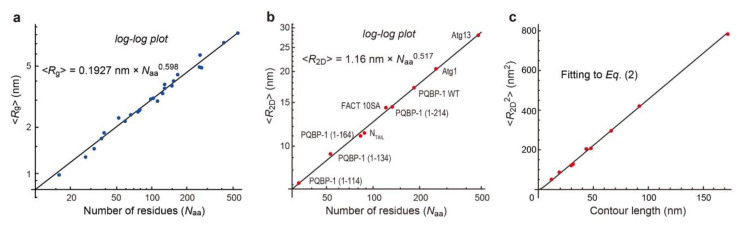
Relationship between molecular dimensions and the number of amino acid residues. (**a**,**b**) Power laws are found in chemically denatured proteins (**a**) and fully disordered regions within IDPs (**b**). The data in (**a**) are from the SAXS study by Kohn et al. [48]. The mean *R*_2D_ values, 〈*R*_2D_〉, in (**b**) are those for fully disordered IDRs observed in the indicated protein constructs. (**c**) The mean square of *R*_2D_ values, 〈*R*_2D_^2^〉, of the nine IDP constructs shown in (**b**) are fitted to Equation (2), from which the value of *P*_2D_ is estimated to be 1.16 nm. The number of residues in (**b**) is those contained in the fully disordered IDRs.

**Figure 3 biomolecules-12-01876-f003:**
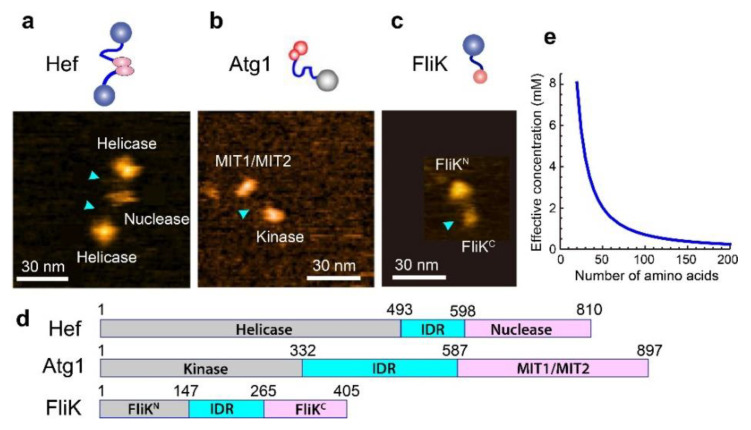
Dumbbell-shaped IDPs observed by HS-AFM. (**a**–**c**) Clips from HS-AFM images and their schematized molecular features of the three IDPs. The light blue arrowheads point to IDRs. (**d**) Domain diagrams of the three IDPs. (**e**) Effective concentration of CTD (NTD) around NTD (CTD) as a function of the number of amino acids contained in the fully disordered IDR connecting NTD and CTD.

**Figure 4 biomolecules-12-01876-f004:**
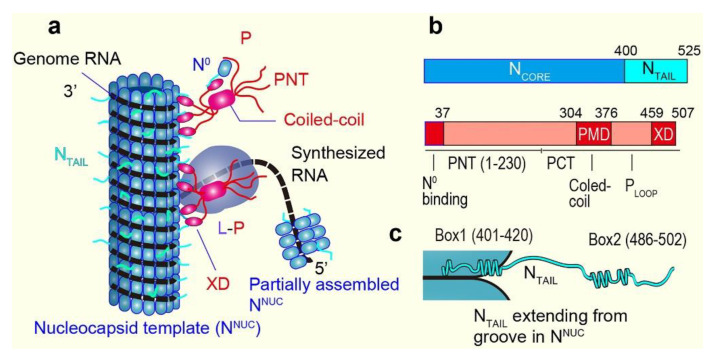
Measles virus IDPs, N and P. (**a**) Schematic showing complexes of N^0^-P, N^NUC^-P, and L-P RNA polymerase. The location of RNA is schematically represented for its visualization. Only a part of N_TAIL_s is shown. (**b**) Domain diagrams of N and P proteins. The light-color regions indicate IDRs. (**c**) Schematic showing N_TAIL_ extending from the groove of nucleocapsid template.

**Figure 5 biomolecules-12-01876-f005:**
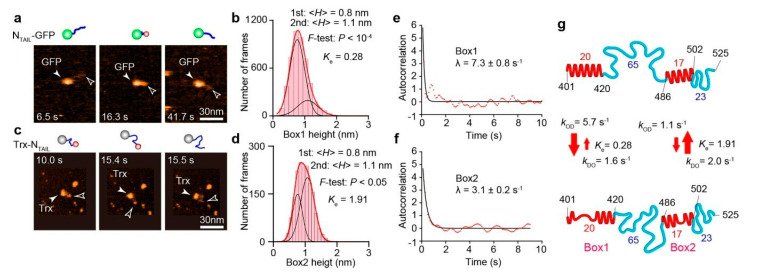
HS-AFM analysis of structure and dynamics of N_TAIL_. (**a**) HS-AFM images and their schematics of N_TAIL_-GFP. The open arrowheads point to the N-terminal end of N_TAIL_. (**b**) Height distribution of the N-terminal globule (i.e., Box1). (**c**) HS-AFM images and their schematics of Trx-N_TAIL_. The open arrowheads point to the C-terminal globule of N_TAIL_. (**d**) Height distribution of the C-terminal globule (i.e., Box2). (**e**,**f**) Autocorrelation functions of time-series height data of Box1 (**e**) and Box2 (**f**). The values of *λ* represent the respective decay constants of the autocorrelation functions. (**g**) Schematics showing structural and dynamic features of N_TAIL_. The top and bottom panels correspond to the more- and less-ordered states, respectively. The red arrows indicate the kinetic nature of changes in height of Box1 and Box2.

**Figure 6 biomolecules-12-01876-f006:**
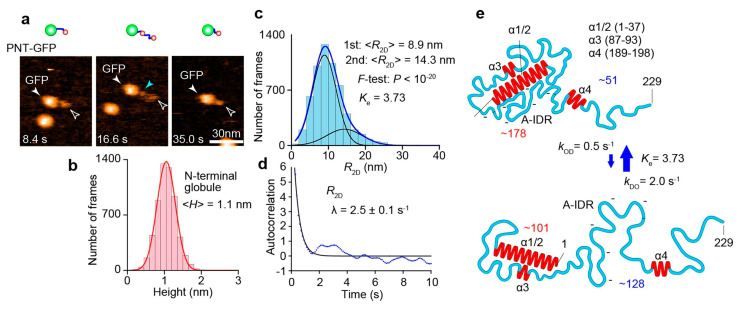
HS-AFM analysis of structure and dynamics of PNT. (**a**) HS-AFM images and their schematics of PNT-GFP. The open arrowheads point to the N-terminal globule of PNT, while the light-blue arrowhead points to a small globule appeared in the middle of disordered region (very likely corresponding to *α*4). (**b**) Height distribution of the N-terminal globule. The value 1.1 nm is identical to the diameter of a single *α*-helix. (**c**) Distribution of end-to-end distance of fully disordered IDR (*R*_2D_). (**d**) Autocorrelation function of time-series data of *R*_2D_. The value of *λ* represents the decay constant of the autocorrelation function. (**e**) Schematics showing structural and dynamic features of PNT. The top and bottom panels correspond to the more- and less-ordered states, respectively. The numbers in red and blue represent the numbers of amino acids contained in the respective globules and fully disordered IDRs, respectively, estimated by using the power law and measured 〈*R*_2D_〉 values. The blue arrows indicate the kinetic nature of changes in *R*_2D_.

**Table 1 biomolecules-12-01876-t001:** Comparison of four techniques used in structural analysis of proteins.

	Data Collection Time	Sample Amount	Spatial Resolution	Dynamics	Molecular Weight limitation
NMR ^(a)^	days–month	2–100 mg	~0.2 nm	ps–ms ^(b)^	<40 kDa
SAXS	s–h ^(c)^	0.1–10 mg	1–10 nm ^(d)^	s-min	none
HS-AFM	~10 s	0.1 μg	2–3 nm (XY) 0.1 nm (Z)	20–100 ms ^(e)^	none
AFM	min	0.1 μg	2–3 nm (XY) 0.1 nm (Z)	>30 s	none

^(a)^ Solution NMR spectroscopy; ^(b)^ Obtainable dynamics do not necessarily include structural coordinates; ^(c)^ Largely depends on the power of X-ray source (long exposure to X-ray damages proteins); ^(d)^ Usually ~5 nm. Revealed molecular shapes are model dependent; ^(e)^ 20–100 ms for IDPs and ~100 ms for structured proteins.

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
