# Peer review of "Functional Implications of Dynamic Structures of Intrinsically Disordered Proteins Revealed by High-Speed AFM Imaging"

_biomolecules, 2022, doi:10.3390/biom12121876_

Round 1
Reviewer 1 Report
This review paper summarizes the state of the art in applying high speed AMF (HS-AFM) to intrinsically disordered proteins (IDP) or intrinsically disordered regions (IDR) within proteins. The paper is well written and clearly organized. After introducing IDP phenomena and discussing the challenges in applying widely used methods of SAXS and NMR, the authors provide a brief introduction to the HS-AFM technology. The experimental approach is sketched, evidence is presented for a lack of major impact from tip contact on the protein. An argument as made for using polymer modeling with planar restrictions on configurations to estimate the number of amino acids in the IDR from the experimental HS-AFM data. Finally examples of HS-AFM studies of several proteins are reviewed with focus on proteins from the measles virus. This review does a great job introducing novice readers to the exciting possibilities of applying HS-AFM to study IDPs and IDRs. It is a strong contribution, and I only offer a couple suggestions for the authors that might improve clarity on some points.
Minor suggestions:
1. When discussing HS-AFM methodology in section 3, it is mentioned that the proteins often have a disordered region extending from a globular domain. The globular domain anchors the protein in place on a surface for repeated scans (as drawn in the schematic of figure 1a). Some readers might wonder what happens if there is no such anchoring domain, and the IDP is free to move on the surface. Could the authors help by adding short mention of what happens in this case? Is this possibly related to the comment later on line 404 in section 6 about the GFP fusion on Ntail impeding otherwise fast diffusion?
2. The paper would benefit from more explicit definitions of the notation used in Section 4 and revisited later in the paper. When introducing <Rg>, mean Rg, and <Rg^2> near lines 190-194 and equation 1, care should be taken to describe intended differences in these quantities. Related, lines 204-214 need more rigorous definition of <R2D> - is it radius of gyration or end-to-end distance? The confusion persists later in the paper with variants of R3D, <R3D^2> <R3D>, R2D, <R2D^2> <R2D>, r, r_sub_zero and R and R_sub_zero appearing throughout sections 5,6 and 7.
3. The question arose for me about exactly what models are represented in equations 1 and 2. It could be mentioned that Equation 1 is often known as the 3-d worm-like chain model (which I believe it is). Is equation 2 the worm-like chain solution in 2D or the end-end distance in 2-D? (2-D wormlike chain radius of gyration is different in: Baschnagel et al. “Semiflexible Chains at Surfaces: Worm-Like Chains and beyond” Polymers 2016, 8, 286; doi:10.3390/polym8080286). The caption for figure 6c seems to indicate that the symbol R2D represents end-to-end distance, but clear definitions earlier in section 4 would be useful. A citation to a derivation of equation 2 would be helpful.
4. What is “N?” on lines 368-369
5. If the images in figure 5a and c are intended as timeseries of HS-AFM, the authors should indicate the time spacing between the panels?
Reviewer 2 Report
The manuscript titled: “Functional Implications of Dynamic Structures of Intrinsically Disordered Proteins Revealed by High-speed AFM imaging” by Prof. Dr. Toshio Ando is a review work where the author display the potential capabilities of high-speed atomic force microscopy (AFM) to image the family of intrinsically disorder proteins (IDPs). The accuracy of this technique combined with its excellent lateral and temporal resolution makes HS-AFM an excellent alternative to other biophysical techniques like cryo-electron microscopy (cry-EM) or X-ray crystallography. Based on the difficulty to achieve reliable information of IDPs due to the flexible dynamic conformation of the protein chains that conform their structure HS-AFM is presented as one main cornerstone to address these properties. The scientific approach and methodology followed by the authors seem right and the showcased information can be relevant for the examined field. The knowledge acquired in the present work could significantly aid in the fields of biomolecular dynamics not only for IDPs systems but also, covering protein complexes in general. The results achieved are well-discussed during the main body of the reported manuscript. The scientific paper is well written. In my opinion the present manuscript is innovative and the methodological approached used matches with the scope of Biomolecules. For the above described reasons, I will recommend the publication in Biomolecules once the following remarks are fixed:
--------
INTRODUCTION
The author provides a narrative introduction depicting the current state-of the art which will aid to the potential readers to better understand the reasons to conduct the present scientific study. Some minor remarks must be addressed in order to improve the quality of this manuscript.
I) “The intrinsically disordered regions (IDRs) contained in IDPs are dynamically protean” (lines 28-29). Please, the author should check this sentence.
II) “However, the resolution of SAXS is low (…)” line 64. Please, the lateral resolution provided by SAXS should be quantified. I strongly recommend to the author to add a Table to summarize all techniques highlighted the main advantages and limitations compared to HS-AFM (information posted in lines 92-93). Of course, lateral and temporal resolutions should appear.
III) Classical AFM imaging measurements of IDPs should be also introduced. In this framework, it exists previous reported work where AFM has successfully reveal the morphology in relevant conditions of tumor suppressor p53 [1] or the non structural protein 3 (NS3) protease from heptatis C virus [2], among others. The author should remark the capabilities of HS-AFM to overcome the limitations presented by classical AFM (Moreover, it may be also convenient to introduce the classical AFM in the Table proposed in point II).
[1] Bizarri, A.R.; et al. A combined atomic force microscopy imaging and docking study to investigate the complex between p53 DNA binding domain and Azurin. J. Mol. Recognit. 2009, 22, 506-515. J. Mol. Recognit. 506-515. https://doi.org/10.1002/jmr.975.
[2] Vega, S.; et al. NS3 protease from hepatitis C virus: biophysical studies on an intrinsically disordered protein domain. Int. J. Mol. Sci. 2013, 14, 13282-13306. https://doi.org/10.3390/ijms140713282.
--------
3. NO EFFECT OF TIP-SAMPLE CONTACT ON THE STRUCTURE OF IDPs
Some points needs to be addressed in this section.
“There may be concern that the physical contact between a protein molecule and an oscillating tip would have a strong impact on the structure and function of the molecule”. Here, the authors should state the type of the exerted tip-sample interactions (lateral forces) and their nature (electrostatic, van der Waals,…). It is interesting to point out that in liquid media these forces are several magnitude orders lower than in air conditions due to the lack of capillarity forces. A brief statement should be add on this regard.
“Note that R2D can be measured accurately (…) an image are not acquired at the same time but at different times during lateral scanning of the sample stage” (lines 168-170). I do not fully understand the information detailed in this sentence. Does the author mean the data acquisition was recorded following the only trace imaging (OTI) mode [3] already known by the author? Some additional details should be provided.
[3] Fukuda, S.; Ando, T. Faster high-speed atomic force microscopy for imaging of biomolecular processes. Rev. Sci. Instrum. 2021, 92, 033705. https://doi.org/10.1063/5.0032948.
--------
4. CONSTANCY OF FLEXIBILITY/RIGIDITY OF FULLY DISORDERED REGIONS
“where R0 = 0.1927 nm (…)” (line 192). “the 0.36 nm” (line 196). “(…) determined to be 0.976 nm” (lines 197-198). Please, the author should homogenize the significant figures. This comment should be taken into account for the rest of the main manuscript body text.
5. IDPs WITH DUMBBELL SHAPE
Figure 3 (line 249). Could it possible to depict the AFM images plotted in panels a-c with the same lateral scale bar? Same comment for Figure 5 (line 415).
--------
9. OUTLOOK
Here, the author points out the promising role that can act HS-AFM to address the morphology of the dynamic structures presented by IDPs. The author should show some potential Industrial applications that this research can significantly contribute. For example, the knowledge gathered could serve to design the next generation of more specific drug compounds against viral infections. Finally, I consider also relevant to state the potential of HS-AFM to address ultrafast unbinding processes of transient biomolecular complexes [5]. This technology can be exploited to overcome the drawbacks of classical AFM measurements on molecular recognition imaging studies [6].
[4] Rizzuti, B.; et al. Design of Inhibitors of the Intrinsically Disordered Protein NUPR1: Balance between Drug Affinity and Target Function. Biomolecules 2021, 11, 1453. https://doi.org/10.3390/biom11101453.
[5] Rico, F.; et al. Heterogeneous and rate-dependent streptavidin-biotin unbinding revealed by high-speed force spectroscopy and atomistic simulations. Proc. Natl. Acad. Sci. U.S.A. 2019, 14, 6594-6601. https://doi.org/10.1073/pnas.1816909116.
[6] Marcuello, C.; et al. Molecular Recognition of Proteins through Quantitative Force Maps at Single Molecule Level. Biomolecules 2022, 12, 594. https://doi.org/10.3390/biom12040594.
--------
REFERENCES
Bibliography citations are in the proper format of Biomolecules. No further actions are required for this section.
--------
OVERVIEW AND FINAL COMMENTS
The submitted work is well-structured and the information shown in the manuscript sections is interesting in the field of intrinsically disordered proteins. Moreover, the author provides a comparative study of IDPs from different biological sources studied by HS-AFM. For this reason, I will recommend the present scientific manuscript for further publication in Biomolecules once all the aforementioned suggestions will be properly fixed.
